# Distractor-Robust Reinforcement Learning via Variational Bisimulation

## Abstract

Model-based reinforcement learning (MBRL) promises data efficiency and generalization, but typical reconstruction-based objectives encourage models to waste representational capacity on task-irrelevant distractors. We introduce VIBES (Variational Inference for Bisimulation-based Encoded States), a new objective that replaces pixel reconstruction with an adversarial term, which enforces that latent states suffice to predict both rewards and the *next* latent state. We show theoretically that, under mild assumptions, global optima of this objective correspond to encoders that induce bisimulation relations, ensuring that latent states capture task-relevant information while discarding irrelevant variation. Our method serves as a drop-in replacement for Dreamer's model-learning component and achieves state-of-the-art performance on the Distracting Control Suite. Unlike prior approaches, it does not rely on image-specific augmentations and applies equally well to high-dimensional vector-state tasks, demonstrated on a 100-link swimmer robot. Finally, latent-space analyses (UMAP embeddings and nearest-neighbor probes) confirm that the learned representations are sensitive to task-relevant structure while invariant to distractors.

## 1 Introduction

Model-based reinforcement learning (MBRL) promises a powerful set of advantages: it can achieve high data efficiency by reusing learned dynamics for many policy updates, support transfer of knowledge across tasks through shared models, enable long-horizon reasoning and counterfactual planning, and, when paired with uncertainty estimates, drive intelligent exploration. These benefits make MBRL an attractive alternative to model-free methods.

However, in practice, learned models are inevitably imperfect—limited by capacity and the cost of collecting data and performing optimization—and this discrepancy between the model and the true environment often leads to suboptimal policies. One might hope that at least model-based RL methods would consistently find the "best" of the available set of imperfect candidate models; however, this is not even the case, because typical model training objectives are not aligned with the ultimate goal of improving control performance. This issue is referred to as objective mismatch (Lambert et al., 2020). Worse still, typical objectives actively incentivize models to expend representational capacity on irrelevant *distractors*—features of the observation that have no bearing on the task.

Consider the case of autonomous driving: while the agent must model the road, traffic signals, and surrounding vehicles, it need not track details such as the positions of leaves on nearby trees. Yet typical model-learning objectives reward the model for reconstructing those leaves just as faithfully as traffic lights, thereby wasting capacity on irrelevant information. This problem has been shown to significantly degrade the performance of otherwise strong algorithms such as Dreamer (Hafner et al., 2020), which performs well on clean image-based benchmarks but struggles in the presence of distractors. This phenomenon exemplifies objective mismatch because the agent actively chooses a model that is poor for policy improvement, even if a model that ignores distractors and is well-suited to policy improvement exists within its model class.

A natural way to mitigate objective mismatch is through state abstraction. State abstraction filters observations so that only task-relevant information is retained. In MBRL, this allows a dynamics model to focus its capacity on predicting what matters for decision-making, rather than reconstructing irrelevant features. Numerous approaches to state abstraction have been proposed, but many

rely on heuristic, domain-specific strategies, such as image augmentations, whose use defeats the purpose of RL in the first place.

In this work, we instead pursue a principled approach based on *bisimulation relations* (Hennessy & Milner, 1985; Givan et al., 2003). Bisimulation provides a theoretical foundation for abstraction by requiring that states be equivalent if and only if they yield identical immediate rewards and identical distributions over future abstract states. This ensures that task-relevant features are preserved while irrelevant variation is discarded. Our work builds on recent approaches that use *bisimulation metrics*—continuous distance functions that approximate behavioral equivalence (Zhang et al., 2021; Castro, 2020). In contrast, we directly learn bisimulation *relations*, showing that our variational objective is optimized only when the encoder induces partitions of histories that satisfy reward and dynamics sufficiency. This yields a representation aligned with the theoretical definition of bisimulation while avoiding the need for brittle and hyperparameter-sensitive distance metrics. We call our resulting model-based RL algorithm **VIBES**: *Variational Inference for Bisimulation-based Encoded States*.

We evaluate our method in the challenging Distracting Control Suite (DCS) (Stone et al., 2021), where standard DeepMind Control Suite tasks are augmented with strong visual distractors. Our approach consistently outperforms prior state-of-the-art RL methods, both model-free and model-based, for distractor robustness, despite requiring no specialized data augmentation. Moreover, we demonstrate that the learned representations generalize to unseen distractors and also yield strong performance in a high-dimensional vector-state control task (a swimmer robot with an $401$-dimensional observation space). Finally, qualitative analyses confirm that our method learns representations that preserve task-relevant structure, discard distractors, and capture symmetries in the dynamics (e.g., rotational invariances), yielding a more compact and general abstraction.

In summary, the contributions of this paper are as follows:

1. We propose a principled state abstraction method based on bisimulation relations, integrated into a variational inference framework for model learning.

2. We demonstrate strong empirical performance on both image-based tasks with heavy distractors (DCS) and high-dimensional vector-state control, outperforming existing approaches without requiring data augmentations.

3. We provide a quantitative analyses showing that our learned representations preserve task-relevant features while discarding irrelevant variation.

## 2 BACKGROUND

### 2.1 VARIATIONAL INFERENCE FOR LEARNING STATE REPRESENTATIONS

provides a general framework for learning latent-variable models. In such models, observations are assumed to be generated from unobserved latent variables through a probabilistic process. Latent-variable models are powerful because they enable compact, structured representations that capture essential features of the data. In the context of RL, Variational inference (VI) enables learning latent state representations together with dynamics and reward models. Dreamer is a prominent VI formulation that maximizes a lower bound on $\log p(o_{0:T}, r_{0:T} \mid a_{0:T-1})$ using a decoder $\hat{p}(o_t \mid s_t)$, a transi-

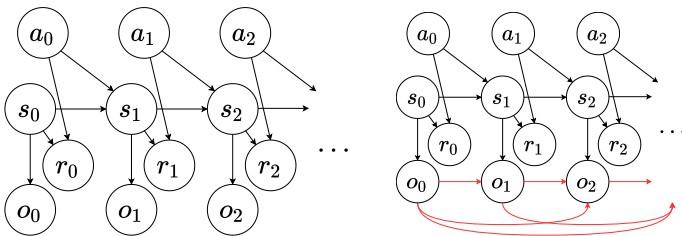

(a) PGM assumed by *Dreamer*.     (b) PGM assumed by *VIBES*.

Figure 1: Side-by-side comparison of graphical model assumptions. **Left (Dreamer):** the observation at time $t$ is conditionally independent of prior observations and actions given the current latent state, i.e., $o_t \perp\!\!\!\perp o_{<t}, a_{<t} \mid \tilde{s}_t$. **Right (Ours):** we allow $o_t$ to depend on the full history of observations (and, in general, on the history of states/actions). Under Dreamer's assumption, any information in $o_{<t}$ needed to predict $o_t$ must be encoded into $s_t$; in our formulation this pressure is removed from $s_t$ because past observations can inform $o_t$ directly.

tion $\hat{p}(s_t \mid s_{t-1}, a_{t-1})$, and a reward model $\hat{p}(r_t \mid s_t, a_t)$. A key modeling assumption is $o_t \perp\!\!\!\perp (o_{<t}, a_{<t}) \mid s_t$, which pressures $s_t$ to carry any history information useful for pixel prediction—even if irrelevant for control (Figure 1a). This "remember everything" incentive misaligns model learning with decision making, especially when in tasks with distractors.

In Sec. 3, we propose a modified objective that relaxes this conditional independence assumption. By allowing observations to depend on past observations directly, we remove the pressure for $\tilde{s}_t$ to transmit irrelevant predictive information, freeing the latent representation to focus on features that matter for predicting rewards and dynamics (Fig. 1b).

## 2.2 BISIMULATION RELATIONS

Bisimulation originates in the theory of Markov decision processes (MDPs) and provides a principled notion of when two states can be considered behaviorally equivalent. Intuitively, two states are bisimilar if they yield the same expected rewards and, under any action, transition to future states that are themselves bisimilar. This recursive definition ensures that bisimilar states are indistinguishable for the purposes of planning and control, making bisimulation relaitons an ideal basis for a state representation.

Formally, let $\mathcal{M} = (\mathcal{S}, \mathcal{A}, P, R, \gamma)$ denote an MDP with state space $\mathcal{S}$, action space $\mathcal{A}$, transition kernel $P(\cdot \mid s, a)$, reward function $R(s, a)$, and discount factor $\gamma \in [0, 1)$. A relation $\sim$ on $\mathcal{S}$ is a *bisimulation relation* if, for all $s_i, s_j \in \mathcal{S}$,

$$s_i \sim s_j \iff \begin{cases} R(s_i, a) = R(s_j, a) & \forall a \in \mathcal{A}, \\ P_{\tilde{s}'|s,a}(\tilde{s}' \mid s_i, a) = P_{\tilde{s}'|s,a}(\tilde{s}' \mid s_j, a) & \forall a \in \mathcal{A}, \quad \forall \tilde{s}' \in \tilde{\mathcal{S}} \end{cases}$$

where $\tilde{\mathcal{S}}$ is the set of all groups of equivalent states, $\tilde{s}'$ is a particular group of equivalent states, and $P_{\tilde{s}'|s,a}(\tilde{s}' \mid s, a) = \sum_{s' \in \tilde{s}'} P(s'|s, a)$ is the probability of transitioning into group $\tilde{s}'$ at the next timestep, given state $s$ and action $a$. Put differently, equivalent states must yield identical immediate rewards under every action and identical distributions over groups of next states. The coarsest such relation partitions the state space into *bisimulation equivalence classes*, which can be treated as abstract states without loss of optimal control performance (Givan et al., 2003; Li et al., 2006).

**Generalizing to Partially Observable MDPs**  When dealing with Partially Observable MDPs (POMDPs), we must replace the *state* of the environment $s$ with the *belief state* $b$, *i.e.,* the posterior belief over the true environmental state given the agent's history. Because the agent history at a given timestep, $H_t = (o_0, a_0, ..., o_t)$ *determines* the belief state $b_t$ through Bayes rule (Castro et al., 2009), we can equivalently think of bisimulation relations as partitioning histories. This is the primary notion of bisimulation that we use in this work.

**Bisimulation for RL**  Bisimulation is appealing for reinforcement learning because it formalizes the idea of *state abstraction*: discarding irrelevant details of the environment while preserving task-relevant information. A representation that encodes bisimulation equivalence classes retains precisely the information needed for reward prediction and decision making. By contrast, states that differ only in irrelevant features (e.g., background distractors) are collapsed into the same abstract state.

Recent work has sought to approximate bisimulation in continuous or high-dimensional settings by learning embeddings that preserve bisimulation distances (Ferns et al., 2004; Zhang et al., 2021). In practice, exact bisimulation is infeasible in complex environments, but the principle provides a powerful guide: representations should be invariant to irrelevant variation while distinguishing states that differ in reward or in their distribution over future behaviors. In this work, we show how to incorporate this principle directly into a variational inference framework for model-based RL, yielding latent representations that approximate bisimulation relations and are robust to distractors.

## 3 LEARNING BISIMULATION RELATIONS WITH VARIATIONAL INFERENCE

We adopt a variational approach to learn a latent state, a dynamics model, and a reward model, but replace pixel reconstruction with an *adversarial* term that measures whether the latent state at time

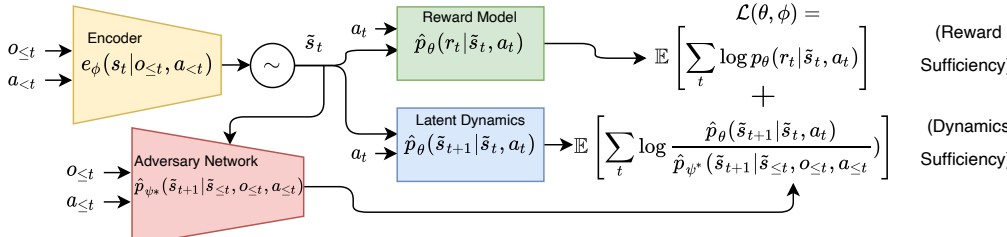

Figure 2: **Model diagram and objective.** Our model comprises an *encoder* (CNN+GRU) that maps history $(o_{\leq t}, a_{<t})$ to a stochastic latent $\tilde{s}_t$; a *dynamics model* that predicts $\tilde{s}_t + 1$ from $(\tilde{s}_t, a_t)$; a *reward model* $p(r_t \mid \tilde{s}_t, a_t)$; and an *adversary network* that predicts $\tilde{s}_{t+1}$ from the history and current action, $(o_{\leq t}, a_{\leq t})$. Our objective has two parts: a *reward sufficiency* term, maximized when the latent state contains all information needed to predict the current reward, and a *dynamics sufficiency* term, maximized when the latent state is as informative as the full history for predicting the *next* latent state.

$t$ is as informative as the full history for predicting the *next* latent state. The resulting objective has two parts: a *reward sufficiency* term (the latent must predict rewards) and a *dynamics sufficiency* term (the latent must be as predictive of the next latent as history). A full derivation is deferred to Appendix A.

### 3.1 MODEL OBJECTIVE

Let $\tilde{s}_t \sim e_\phi(o_{\leq t}, a_{<t})$ denote the stochastic latent at time $t$, where $e_\phi$ is an *encoder network*. In practice, our encoder outputs a Guassian with fixed variance 0.01, and a mean with an $l_2$ norm of 1, thus fixing an upper bound on the amount of information that can be encoded into $\tilde{s}_t$.

Our model objective is

$$\mathcal{L}(\theta, \phi) \;=\; \mathbb{E}\left[\sum_t \log \frac{\hat{p}_\theta(\tilde{s}_t \mid \tilde{s}_{t-1}, a_{t-1})}{\hat{p}_{\psi^*}(\tilde{s}_t \mid \tilde{s}_{<t}, o_{<t}, a_{<t})} \;+\; \log \hat{p}_\theta(r_t \mid \tilde{s}_t, a_t)\right]. \tag{1}$$

Here $\hat{p}_\theta(\tilde{s}_t \mid \tilde{s}_{t-1}, a_{t-1})$ is the latent dynamics; $\hat{p}_\theta(r_t \mid \tilde{s}_t, a_t)$ is the reward model; and $\hat{p}_{\psi^*}(\tilde{s}_t \mid \tilde{s}_{<t}, o_{<t}, a_{<t})$ is an *adversary* (the optimal history-conditioned predictor of $\tilde{s}_t$), obtained by an inner maximization (see App. A).

**Intuitive explanation of $\mathcal{L}$.** The reward sufficiency term $\log p(r_t|\tilde{s}_t, a_t)$ encourages $\tilde{s}_t$ to be sufficient for predicting $r_t$ given $a_t$. The dynamics sufficiency term $\frac{\hat{p}_\theta(\tilde{s}_t|\tilde{s}_{t-1}, a_{t-1})}{\hat{p}_{\psi^*}(\tilde{s}_t|\tilde{s}_{<t}, o_{<t}, a_{<t})}$ compares two predictors of $\tilde{s}_t$: our latent dynamics model $\hat{p}_\theta(\tilde{s}_t|\tilde{s}_{t-1}, a_{t-1})$ that only sees $(\tilde{s}_{t-1}, a_{t-1})$ vs. an optimal predictor that sees the full history, $\hat{p}_{\psi^*}(\tilde{s}_t|\tilde{s}_{<t}, o_{<t}, a_{<t})$. This log ratio is upper-bounded by 0 and equals 0 iff $\tilde{s}_{t-1}$ contains all history information needed to predict $\tilde{s}_t$. Maximizing equation 1 therefore pushes the representation toward *reward* and *dynamics* sufficiency.

### 3.2 WHY $\mathcal{L}$ YIELDS BISIMULATION

We formalize the guarantee with a lemma (sufficiency $\Rightarrow$ bisimulation) and a theorem (optimality of $\mathcal{L} \Rightarrow$ sufficiency). Full proofs are in App. B.

**Lemma 3.1** (Sufficiency induces bisimulation). *Let $\tilde{s}_t = e_\phi(o_{\leq t}, a_{<t})$. Suppose for all $t$ and actions $a_t$:*

$$(\textit{Dynamics sufficiency}) \quad \tilde{s}_t \perp\!\!\!\perp (o_{<t}, a_{<t-1}) \mid \tilde{s}_{t-1}, a_{t-1}, \tag{2}$$

$$(\textit{Reward sufficiency}) \quad r_t \perp\!\!\!\perp (o_{\leq t}, a_{<t}) \mid \tilde{s}_t, a_t. \tag{3}$$

*Then the history partition induced by $e_\phi$ is a bisimulation relation.*

**Theorem 3.2** (Global optima of $\mathcal{L}$ yield bisimulation). *Assume sufficiently expressive model classes for rewards and dynamics. Any global maximizer of equation 1 satisfies, for all $t$,*

$$\hat{p}_\theta(\tilde{s}_t \mid \tilde{s}_{t-1}, a_{t-1}) = p(\tilde{s}_t \mid \tilde{s}_{<t}, o_{<t}, a_{<t}), \qquad \hat{p}_\theta(r_t \mid \tilde{s}_t, a_t) = p(r_t \mid o_{\leq t}, a_{\leq t}).$$

*These equalities imply the sufficiency conditions in Lemma 3.1, hence the encoder-induced partition is a bisimulation.*

*Proof Sketch.* By Jensen's inequality, $\mathcal{L}$ is maximized if and only if

$$\hat{p}_\theta(\tilde{s}_t \mid \tilde{s}_{t-1}, a_{t-1}) = p(\tilde{s}_t \mid \tilde{s}_{<t}, o_{<t}, a_{<t}), \qquad \hat{p}_\theta(r_t \mid \tilde{s}_t, a_t) = p(r_t \mid o_{\leq t}, a_t)$$

which implies that $e_\phi$ satisfies equation 2 and equation 3, which by equation 3.1 implies that $e_\phi$ is a bisimulation relation. A complete proof is provided in Appendix B.2.

**Optimizing the Model Objective**    Optimizing our model objective in equation 1 is slightly more involved than in Dreamer because it contains an inner maximization problem (equation 9) used to compute $\psi^*$. Fortunately, as we prove in Appendix B.3, the derivative with respect to the explicit occurrence of $\phi$ in the adversarial objective vanishes, *i.e.*, $\frac{\partial}{\partial \phi} \mathbb{E}[\log \hat{p}_{\psi^*}(\tilde{s}_t \mid \tilde{s}_{<t}, o_{<t}, a_{<t})] = 0$, allowing us to treat $\psi^*$ as fixed when computing $\nabla_\theta \mathcal{L}$ and $\nabla_\phi \mathcal{L}$. This allows the model training to be reduced from a bilevel optimization problem, to a simpler alternating optimization, where $\psi$ is optimized before each gradient update of $\theta$ and $\phi$. In practice, we found it sufficient to perform one step of gradient ascent on $\psi$ for each step of gradient ascent on $\theta$ and $\phi$. Our complete model training procedure is summarized in Algorithm 1 in Appendix C.1.

**Integration into Online Reinforcement Learning**    The objective in equation 1 serves as a direct drop-in replacement for the model-learning component of Dreamer. Once trained, the resulting world model is used in a manner identical to Dreamer: to simulate imagined rollouts in the latent state space for policy and value learning. Thus, apart from replacing the reconstruction loss with our bisimulation-driven objective, the rest of the Dreamer framework (policy optimization, value estimation, and training loop) remains unchanged.

## 4 EXPERIMENTS

We design our experiments to evaluate three key questions: (1) Does our approach improve robustness to distractors in challenging image-based control tasks? (2) How does each component of our objective contribute to performance? (3) Do the learned representations capture task-relevant structure in a way that generalizes beyond pixel observations?

To this end, we first benchmark our method on the Distracting Control Suite (DCS), which provides a standardized and challenging setting for evaluating control from images in the presence of visual distractions. We then conduct ablation studies to isolate the effect of the adversarial objective. To test generality, we further evaluate on high-dimensional continuous control tasks from vector states. Finally, we analyze the learned latent spaces to qualitatively assess whether they preserve task-relevant features while discarding irrelevant variation.

### 4.1 DISTRACTING CONTROL SUITE

We first evaluate our method on the Distracting Control Suite (DCS) (Stone et al., 2021), a benchmark explicitly designed to test robustness to visual distractors. The DCS extends the DeepMind Control Suite with three types of distractors: (i) *background distractors*, where high-resolution natural videos play behind the agent, (ii) *color distractors*, where the agent's body color changes dynamically, and (iii) *camera distractors*, where the viewpoint of the camera shifts during the episode. Together, these settings create a challenging testbed for evaluating algorithms that must learn robust control policies from pixel observations while ignoring irrelevant features of the input.

This benchmark is particularly well suited to our setting: while solving each task requires fine-grained control over the agent's state (e.g., catching a ball, swinging up a cartpole), the distractors introduce large amounts of irrelevant variation that should ideally be filtered out by the learned representation. In this sense, the DCS provides a direct test of whether an algorithm can overcome the objective mismatch problem discussed in Section 1. We compare our algorithm against a suite of strong baselines, including SAC+RAD (Laskin et al., 2020b), QT-Opt variants with RAD and DrQ regularization, RSAC (Gupta et al., 2022), CURL (Laskin et al., 2020a), CoRe (Srivastava et al.), and SAR (Liang et al., 2024). All methods are evaluated for 500K environment steps, following

| Method | Mean | BiC-Catch | C-swingup | C-run | F-spin | R-easy | W-walk |
|---|---|---|---|---|---|---|---|
| SAC+RAD† | $270 \pm 31$ | $366 \pm 59$ | $297 \pm 21$ | $198 \pm 39$ | $338 \pm 59$ | $173 \pm 11$ | $249 \pm 138$ |
| QT-Opt+RAD† | $343 \pm 24$ | $490 \pm 64$ | $467 \pm 12$ | $170 \pm 8$ | $393 \pm 91$ | $428 \pm 68$ | $109 \pm 12$ |
| QT-Opt+DrQ† | $265 \pm 5$ | $395 \pm 39$ | $431 \pm 18$ | $126 \pm 10$ | $203 \pm 33$ | $343 \pm 53$ | $91 \pm 3$ |
| RSAC† | $275 \pm 24$ | $181 \pm 32$ | $465 \pm 21$ | $292 \pm 10$ | $86 \pm 55$ | $145 \pm 31$ | $482 \pm 20$ |
| CURL† | $391 \pm 30$ | $102 \pm 20$ | $432 \pm 15$ | $233 \pm 13$ | $648 \pm 32$ | $253 \pm 40$ | $\underline{678 \pm 35}$ |
| CoRe† | $586 \pm 30$ | $\underline{798 \pm 30}$ | $499 \pm 22$ | $423 \pm 22$ | $\mathbf{713 \pm 81}$ | $340 \pm 60$ | $\mathbf{744 \pm 40}$ |
| SAR | $173 \pm 17$ | $286 \pm 84$ | $351 \pm 29$ | $240 \pm 70$ | $14 \pm 16$ | $108 \pm 31$ | $36 \pm 3$ |
| VIBES (Ours) | $\mathbf{701 \pm 29}$ | $\mathbf{844 \pm 95}$ | $\mathbf{696 \pm 50}$ | $\mathbf{489 \pm 22}$ | $670 \pm 167$ | $\mathbf{930 \pm 12}$ | $574 \pm 56$ |
| No Adv. Obj. | $606 \pm 27$ | $593 \pm 158$ | $\underline{658 \pm 28}$ | $436 \pm 27$ | $469 \pm 57$ | $\underline{925 \pm 21}$ | $556 \pm 23$ |
| Recon. Obj. | $417 \pm 49$ | $144 \pm 96$ | $543 \pm 72$ | $\underline{469 \pm 11}$ | $347 \pm 164$ | $360 \pm 280$ | $\underline{639 \pm 101}$ |

Table 1: Performance comparison across Distracting Control Suite tasks at 500K environment steps. We report mean episode return $\pm$ 95% confidence interval across 5 seeds. Bolded numbers indicate the highest mean return in each column. Underlined numbers indicate results whose confidence intervals overlap with the bolded entry, suggesting statistically indistinguishable performance at the reported level. Our method (VIBES) achieves the highest average performance across tasks and the best final return in four out of six environments, and is the only algorithm to solve *Reacher-easy*. The "No Adv. Loss" ablation confirms the importance of the adversarial term: performance drops across tasks when it is removed, but remains the second-best overall method. † denotes results taken from prior work.

standard protocol. All evaluations use a *held-out set of background video distractors*, directly testing the generalization of the learned state representation.

As shown in Table 1, our approach achieves the highest mean performance across tasks and the best average final return in four out of six environments. Notably, our method is the only one to successfully solve the *Reacher-easy* task, achieving a near-perfect score. This result is particularly interesting because Reacher-easy is structurally simpler than other DCS tasks, yet most prior methods fail on it. We hypothesize that this difference arises from the multi-goal nature of Reacher-easy, where the goal location is randomized each episode. Many existing algorithms incentivize representations that primarily encode agent-centric dynamics—i.e., the aspects of the system directly influenced by actions. Since the goal position is unaffected by the agent's actions, it is often underrepresented in the learned latent state, even though it is critical for solving the task. In contrast, our method explicitly enforces preservation of reward-predictive information through bisimulation relations, ensuring that both agent state and goal location are encoded. This explains both the robustness of our approach across distractor types and its strong relative advantage on multi-goal settings like Reacher-easy.

**Ablation Studies** To better understand the contribution of individual components of our objective, we perform ablation studies on the DCS tasks. We first remove the adversarial objective, retaining only the dynamics and reward reconstruction objectives. As shown in Table 1, this ablation leads to a drop in performance across all tasks, confirming that the adversarial term provides a measurable benefit. However, the ablation remains the second-best performing method overall, outperforming all existing baselines despite the removal of this loss term.

We hypothesize that performance remains strong in this ablation because in environments where the underlying bisimulation state (i.e., the minimal task-relevant representation) is (1) governed by deterministic dynamics and (2) fully observable from the raw observations, the adversarial objective becomes redundant. Under these conditions, both $e_\phi$ and $p(\tilde{s}_t|\tilde{s}_{<t}, o_{<t}, a_{<t})$ become deterministic. The adversarial objective therefore reduces to a constant and can be omitted without changing the optimal solution. Condition (1) holds in our setting because the underlying MuJoCo simulator is deterministic. Condition (2), however, is not guaranteed: information about the true agent state may be obscured due to limited rendering resolution or ambiguity introduced by distractors (e.g., when background motion or camera changes overlap with agent movement). The adversarial term is therefore most valuable in precisely those cases where observations are imperfectly informative about the underlying state, which explains its empirical benefit.

We also evaluate an ablation in which we retain the same model architecture but replace our adversarial objective with a Dreamer-style pixel reconstruction objective:

$$\mathcal{L}_{\text{recon}}(\theta, \phi) = \mathbb{E}_{\vec{o}, \vec{r}, \vec{a}, \vec{s}} \left[ \sum_t \left( \log \hat{p}_\theta(o_t | \tilde{s}_t) + \log \hat{p}_\theta(r_t | \tilde{s}_t, a_t) + \log \frac{\hat{p}_\theta(\tilde{s}_t | \tilde{s}_{t-1}, a_{t-1})}{e_\phi(\tilde{s}_t | o_{\leq t}, a_{< t})} \right) \right] . \text{[1]}$$

This ablation isolates the contribution of our objective from that of the architecture. Results are reported in Table 1. Overall, replacing the adversarial loss with reconstruction leads to a noticeable drop in performance. Interestingly, in certain tasks such as *Cheetah-run* and *Walker-walk*, performance was only marginally affected, or even improved. Understanding the conditions under which reconstruction-based objectives perform competitively remains an open question for future work.

## 4.2 HIGH-DIMENSIONAL CONTINUOUS CONTROL FROM VECTOR STATES

To evaluate the generality of our method beyond image-based tasks, we study a high-dimensional continuous control problem based on the *Swimmer* environment from the DeepMind Control Suite (Tassa et al., 2018). The standard Swimmer task involves a procedurally generated $k$-link planar snake robot moving in a viscous fluid. For our experiments, we extend this environment to a 100-link swimmer, resulting in a state dimension of 204 and an observation dimension of 401. The goal of the agent is to propel its nose to a fixed target location, with a reward that decreases smoothly with distance according to a Lorentzian function.

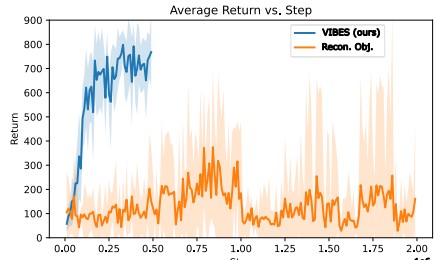

Figure 3: **100-link swimmer (vector states): learning curves.** Episode returns versus environment steps on the 100-link swimmer task with 401-dimensional vector observations. Our bisimulation-driven objective achieves higher and more stable performance than a Dreamer-style reconstruction baseline with identical architectures.

Directly controlling a 99-dimensional action space is impractical, and not the focus of our study. Instead, we parameterize the action space in terms of a spatiotemporal sinusoid that governs joint torques. Specifically, the torque applied to joint $i$ at time $t$ is given by $\tau_i(t) = A \cdot \sin(c_1 i + c_2 t) + c_3$, where $i$ is the link index and $t$ is the timestep. The agent outputs the four parameters $[A, c_1, c_2, c_3]$, which are scaled to lie within a reasonable range for stable control. We additionally use an action repeat of 10 to prevent the agent from changing sinusoid parameters at unrealistically high frequencies.

This setting serves two purposes. First, it provides an extremely high-dimensional state space (204 dimensions), testing whether our approach can scale to vector-state control problems where direct reconstruction is infeasible. Second, it highlights that our method is not tied to image-specific inductive biases: unlike prior approaches that rely on augmentations such as random crops or color jittering, our objective applies equally well to arbitrary input domains.

We compare our algorithm against a reconstruction-based baseline modeled after Dreamer, using identical encoder, dynamics, and reward architectures for a controlled comparison. Across multiple runs, we find that our method learns more stable policies and achieves significantly higher returns, while the Dreamer-style reconstruction objective struggles to make progress in the complex high-dimensional state space. These results reinforce our claim that bisimulation-based objectives offer a principled and general approach to state abstraction, applicable to both visual and non-visual domains.

## 4.3 LATENT SPACE ANALYSIS

To probe whether the learned representation preserves task-relevant structure while discarding distractors, we analyze UMAP embeddings of state representations on the *Reacher-easy* task. We generate 10 unique trajectories by rolling out a trained policy from 10 distinct initial states, and pair each with 10 unique background distractor trajectories, forming a $10 \times 10$ grid of combinations.

---

[1]Because our encoder uses a fixed variance, the term in gray is constant and therefore omitted during optimization.

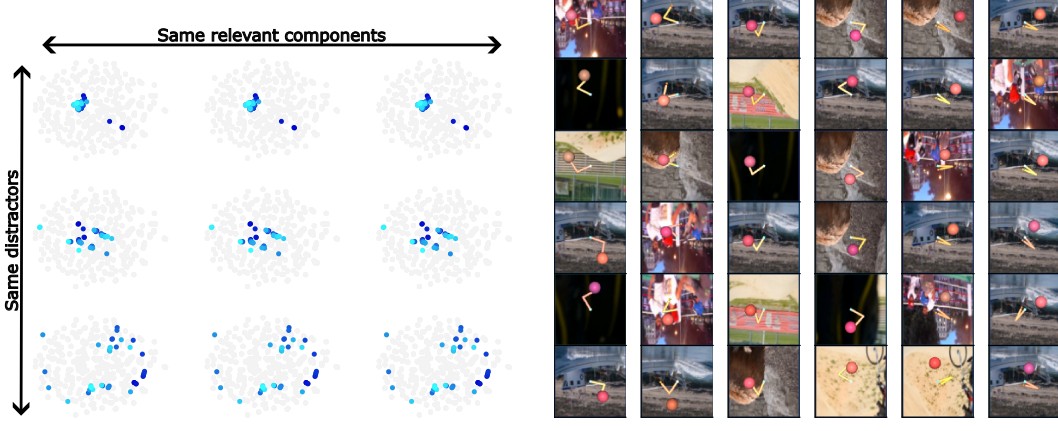

(a) UMAP embeddings (3×3).    (b) Nearest neighbors in latent space.

Figure 4: **Latent-space analyses on *Reacher-easy*.** (a) We visualize 2D UMAP projections of 9 encoded trajectories in a 3×3 grid, where rows share relevant states while columns share distractors. Variation across rows is large but small within rows, indicating that the encoder is sensitive to relevant states while invariant to distractors. (b) For six observations (top of each column), we show their five nearest neighbors in the latent space. The encoder groups together observations with different distractors and reveals that the model maps rotationally symmetric states with equivalent arm–goal configurations to nearby representations.

Each trajectory is encoded by our trained encoder to produce latent state representations, which are then mapped to 2D with UMAP (McInnes et al., 2018).

Figure 4a shows a $4 \times 4$ array of scatter plots. Each panel depicts the embeddings from *all* timesteps across all 100 trajectories in gray, while highlighting a single trajectory in blue. Panels in the same *row* correspond to trajectories with the same relevant states but different distractors; panels in the same *column* share distractors while differing in relevant states.

We observe substantial variation across rows but comparatively little variation within a row. This indicates that the latent representation is sensitive to changes in the underlying relevant state, while remaining largely invariant to distractors—consistent with bisimulation-driven abstraction.

While the latent space analysis in Figure 4a confirms that the learned representations are insensitive to distractors, it does not reveal which aspects of the relevant state are preserved versus collapsed. To probe this question, we conducted a nearest-neighbor analysis in latent space. Using a trained encoder, we encoded the entire dataset generated during a training run of online RL in the *Reacher-easy* environment. From this dataset, we selected 20 random latent states and, for each, identified the five most similar latent states from other trajectories. Similarity was measured using *standardized L2 distance*, which normalizes each dimension by its variance to avoid dominance by high-variance components. Of the 20 randomly selected states, we visualize six representative cases in Figure 4b, chosen to include a mixture of on-goal and off-goal states. The full set of 20 cases is visualized in Appendix Figure 5.

The analysis supports two main findings. First, as expected, the encoder is largely invariant to distractors: states with different backgrounds and agent colors are grouped together in latent space. Second, the encoder appears to have learned a rotational symmetry of the Reacher problem: because the optimal action sequence is unchanged by jointly rotating the arm and goal location about the origin, the encoder clusters together configurations that are equivalent under such rotations. This suggests that the learned representation is even more compact than the true underlying vector state, which preserves absolute angles. These results indicate that the encoder captures task-relevant structure while discarding both distractors and redundant symmetries, consistent with the bisimulation perspective.

## 5 Conclusions

We introduced **VIBES** (Variational Inference for Bisimulation-based Encoded States), a model-learning objective that replaces pixel reconstruction with two control-aligned terms: a *reward sufficiency* term and a *dynamics sufficiency* term. We proved that, under mild expressivity assumptions, global optima of VIBES correspond to encoders that induce bisimulation relations on histories, ensuring that the learned representation preserves task-relevant structure while collapsing distractors. Empirically, VIBES serves as a drop-in replacement for Dreamer's model-learning component and achieves state-of-the-art performance on the Distracting Control Suite without image-specific augmentations. We further demonstrated applicability to high-dimensional vector-state control and provided qualitative evidence that the learned latents are invariant to distractors and capture symmetries of the underlying dynamics. Overall, VIBES advances representation learning for MBRL by bridging the gap between the theoretically appealing framework bisimulation relations and empirically successful and robust framework of VI.

## 6 Limitations and Future Work

While promising, our approach has several limitations that suggest fruitful directions for future research:

1. **Approximation to bisimulation.** Our guarantees hold at global optima with sufficiently expressive model classes. In practice, limited capacity, finite data, and imperfect optimization mean we cannot quantify how close the learned representation is to a true bisimulation relation.

2. **Choice of abstraction.** Bisimulation is not the only task-aligned notion of state abstraction. Alternatives—e.g., value-equivalence, successor features, predictive state representations, or information-bottleneck criteria—may be preferable for certain objectives (e.g., policy transfer or exploration). Extending VIBES to interpolate among these notions is an open direction.

3. **Computation.** The adversary adds compute and memory overhead compared to pure reconstruction-free objectives.

4. **Scope of evaluation.** We evaluate on DCS and one high-dimensional vector-state task. Validation on real robots, harder exploration regimes (sparse rewards), long-horizon tasks, and domains with more realistic distractors remains for future work.

5. **Design choices in the bottleneck.** Our fixed-variance, unit-norm latent parameterization imposes a specific information bottleneck that may interact with task difficulty. Learning the bottleneck (e.g., via learnable variance or rate–distortion trade-offs) could improve adaptability across domains.

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

## A    DERIVATION OF THE MODEL OBJECTIVE

**Notation**    Throughout the derivation, we use $\vec{o}$, $\vec{a}$, $\vec{r}$, and $\vec{\tilde{s}}$ to denote temporal sequences of observations, actions, rewards, and state representations, respectively. We use $p(\cdot)$ to denote *true* probability distributions (*i.e., those specified by the environment*) and $\hat{p}(\cdot)$ to denote approximations to said true distributions (*i.e.,* those specified by our model). We use $\tilde{s}_t \sim e_\phi(\cdot|o_{\leq t}, a_{<t})$ to denote the distribution over the state representation at time $t$, which depends on the observations before and including time $t$, and the actions up to time $t$, and is parameterized by $\phi$. Note that we do not place a circumflex ( ˆ ) over $e_\phi$, because this is our *chosen* state representation and does not represent an approximation to some underlying true distribution.

Similar to our derivation of the Dreamer model objective, we start by lower-bounding the expected log ratio of the approximate and true distributions over our observations and rewards given actions, according to

$$\mathbb{E}_{\vec{o},\vec{r},\vec{a} \sim D} \left[ \log \frac{\hat{p}(\vec{o},\vec{r}|\vec{a})}{p(\vec{o},\vec{r}|\vec{a})} \right] \geq \mathbb{E}_{\vec{o},\vec{r},\vec{a} \sim D} \mathbb{E}_{\vec{\tilde{s}} \sim e_\phi} \left[ \log \frac{\hat{p}(\vec{o},\vec{r},\vec{\tilde{s}}|\vec{a})}{e_\phi(\vec{\tilde{s}}|\vec{o},\vec{a})p(\vec{o},\vec{r}|\vec{a})} \right]. \tag{4}$$

From here, we diverge from the Dreamer derivation, factoring both the numerator and denominator differently.

**Numerator** Inthe Dreamer derivation, $\hat{p}(\vec{o},\vec{r},\vec{\tilde{s}}|\vec{a})$ factors into an *observation decoder* $\hat{p}_\theta(o_t|\tilde{s}_t)$,*latent dynamics model* $\hat{p}_\theta(\tilde{s}_t|\tilde{s}_{t-1},a_{t-1})$, and a *reward decoder* $\hat{p}_\theta(r_t|\tilde{s}_t,a_t)$. We instead factor the numerator according to

$$\hat{p}(\vec{o},\vec{r},\vec{\tilde{s}}|\vec{a}) = \prod_{t=1}^{T} {\color{red} p(o_t|\tilde{s}_{\leq t}, o_{<t}, a_{<t})} \hat{p}_\theta(\tilde{s}_t|\tilde{s}_{t-1},a_t)\hat{p}_\theta(r_t|\tilde{s}_t,a_t). \tag{5}$$

Note that the term in red has two key differences from the corresponding term in Dreamer. First, it conditions on $o_{<t}$, thereby avoiding the assumption that the system is Markovian in $s$. This allows us to avoid learning a state representation that is incentivized to encode all (potentially irrelevant) information from past observations that is predictive for the current observation (see Figure 1 right for a visual representation of these assumptions). Second, we use the *true* (but unknown) distribution over $p(o_t|\tilde{s}_{\leq t}, o_{<t}, a_{<t})$ rather than a model-approximated distribution, allowing us to avoid having to learn an observation decoder and the associated challenges of representing high-dimensional probability distributions. As we shall see, the fact that this distribution is unknown will not impact our ability to optimize the resulting model objective because this term will cancel.

**Denominator** In the Dreamer derivation, $p(\vec{o},\vec{r}|\vec{a})$ is treated as a constant and eliminated. We instead chose to group it together with the encoder distribution term and re-factor it. Grouping with the encoder term gives us the *true joint distribution* over $\vec{o}$, $\vec{r}$, $\vec{\tilde{s}}$ given $\vec{a}$ according to $e_\phi(\vec{\tilde{s}}|\vec{o},\vec{a})p(\vec{o},\vec{r}|\vec{a}) = p(\vec{o},\vec{r},\vec{\tilde{s}}|\vec{a})$, which can be factored as

$$p(\vec{o},\vec{r},\vec{\tilde{s}}|\vec{a}) = \prod_{t=1}^{T} p(o_t|\tilde{s}_{\leq t}, o_{<t}, a_{<t})p(\tilde{s}_t|\tilde{s}_{<t}, o_{<t}, a_{<t})p(r_t|o_{\leq t}, a_{\leq t}). \tag{6}$$

Substituting equation 5 and equation 6 back into equation 4 gives us

$$\mathbb{E} \left[ \log \frac{\hat{p}(\vec{o},\vec{r}|\vec{a})}{p(\vec{o},\vec{r}|\vec{a})} \right] \geq \mathbb{E} \left[ \log \frac{\prod_{t=1}^{T} \cancel{p(o_t|\tilde{s}_{\leq t}, o_{<t}, a_{\leq t})}\hat{p}_\theta(\tilde{s}_t|\tilde{s}_{t-1},a_{t-1})\hat{p}_\theta(r_t|\tilde{s}_t,a_t)}{\prod_{t=1}^{T} \cancel{p(o_t|\tilde{s}_{\leq t}, o_{<t}, a_{<t})}p(\tilde{s}_t|\tilde{s}_{<t}, o_{<t}, a_{<t})p(r_t|o_{\leq t}, a_{\leq t})} \right] \tag{7}$$

$$= \mathbb{E} \left[ \sum \log \frac{\hat{p}_\theta(\tilde{s}_t|\tilde{s}_{t-1},a_{t-1})}{p(\tilde{s}_t|\tilde{s}_{<t}, o_{<t}, a_{<t})} + \log \hat{p}_\theta(r_t|\tilde{s}_t,a_t) \right] + \mathcal{H}(p(\vec{r}|\vec{o},\vec{a})). \tag{8}$$

Because $\mathcal{H}(p(\vec{r}|\vec{o},\vec{a}))$ does not depend on our model parameters, it can be treated as a constant and ignored. $p(\tilde{s}_t|\tilde{s}_{<t}, o_{<t}, a_{<t})$ is unknown; however, for a sufficiently flexible class of function approximators $\Psi$, by Gibbs' inequality, it is given by

$$p(\tilde{s}_t|\tilde{s}_{<t}, o_{<t}, a_{<t}) = \hat{p}_{\psi*}(\tilde{s}_t|\tilde{s}_{<t}, o_{<t}, a_{<t}) = \underset{\psi \in \Psi}{\text{argmax}} \mathbb{E} \left[ \log \hat{p}_\psi(\tilde{s}_t|\tilde{s}_{<t}, o_{<t}, a_{<t}) \right]. \tag{9}$$

Substituting this term into equation 8 yields our final model objective,

$$\mathcal{L}(\theta,\phi) = \mathbb{E} \left[ \sum \log \frac{\hat{p}_\theta(\tilde{s}_t|\tilde{s}_{t-1},a_{t-1})}{\hat{p}_{\psi*}(\tilde{s}_t|\tilde{s}_{<t}, o_{<t}, a_{<t})} + \log \hat{p}_\theta(r_t|\tilde{s}_t,a_t) \right]. \tag{10}$$

We refer to $\hat{p}_{\psi*}$ as the *adversary network* because optimizing it *reduces* $\mathcal{L}$, placing $\psi$ in an adversarial game with $\phi$ and $\theta$. Similarly, we refer to $\mathbb{E} \left[ \sum_t \log \frac{1}{\hat{p}_{\psi*}(\tilde{s}_t|\tilde{s}_{<t}, o_{<t}, a_{<t})} \right]$ as the *adversarial*

*objective component.* In practice, to ensure that the adversary network never underperforms the dynamics model in predicting $\tilde{s}_t$, we clip each per-dimension term $\mathbb{E}\left[\log \frac{\hat{p}_\theta(\tilde{s}_t^{(i)}|\tilde{s}_{t-1}, a_{t-1})}{\hat{p}_{\psi*}(\tilde{s}_t^{(i)}|\tilde{s}_{<t}, o_{<t}, a_{<t})}\right]$ to be non-positive. This modification does not alter the theoretical objective, since the term is upper-bounded by zero, but it improves stability in practice when the adversary is slow to converge.

# B   PROOFS

## B.1   PROOF OF LEMMA 3.1

Here we restate lemma 3.1 more rigorously:

**Lemma 3.1** (Encoder-induced equivalence and bisimulation).

*Let $H_t = (o_0, a_0, \ldots, a_{t-1}, o_t)$ denote the history of observations and actions up to time $t$. Let $\tilde{s}_t = e_\phi(H_t)$ be the abstract state produced by an encoder $e_\phi$. The encoder induces an equivalence relation $\sim_{e_\phi}$ on histories, where*

$$H_t \sim_{e_\phi} H_t' \quad \Longleftrightarrow \quad e_\phi(H_t) = e_\phi(H_t').$$

*Suppose $e_\phi$ satisfies the following sufficiency conditions for all $t$ and actions $a_t$:*

$$\textbf{(Dynamics sufficiency)} \quad \tilde{s}_{t+1} \perp\!\!\!\perp H_t \mid \tilde{s}_t, a_t, \tag{11}$$
$$\textbf{(Reward sufficiency)} \quad r_t \perp\!\!\!\perp H_t \mid \tilde{s}_t, a_t. \tag{12}$$

*Then the equivalence relation $\sim_{e_\phi}$ is a bisimulation relation on histories, i.e., for all $H_t \sim_{e_\phi} H_t'$,*

$$P_{\tilde{s}'|H,a}(\tilde{s}_{t+1}|H_t, a_t) = P_{\tilde{s}'|H,a}(\tilde{s}_{t+1}|H_t', a_t) \quad \forall a_t \in \mathcal{A}, \quad \tilde{s}_{t+1} \in \tilde{S}, \tag{13}$$
$$P_{r|H,a}(r_t|H_t, a_t) = P_{r|H,a}(r_t|H_t', a_t) \quad \forall a_t \in \mathcal{A}, \tag{14}$$

where $P_{r|H,a}$ is the reward distribution. Note that, while classical bisimulation is often defined in terms of deterministic reward functions $R(H, a)$, this is equivalent to requiring equality of the degenerate reward distribution $P_{r|H,a}(r|H, a) = \delta(r - R(H, a))$. Our formulation in terms of $P_{r|H,a}$ generalizes this definition to stochastic rewards and is thus more naturally aligned with the probabilistic framework of variational inference.

*Proof*: For all $H_t \sim_{e_\phi} H_t'$, for all $\tilde{s}_{t+1} \in \tilde{\mathcal{S}}$, and for all $a_t \in \mathcal{A}$, we have

$$P(\tilde{s}_{t+1}|H_t, a_t) = P(\tilde{s}_{t+1}|H_t, e_\phi(H_t), a_t) \quad \text{(Because } \tilde{s}_{t+1} \perp\!\!\!\perp e_\phi(H_t)|H_t) \tag{15}$$
$$= P(\tilde{s}_{t+1}|e_\phi(H_t), a_t) \quad \text{(By equation 11)} \tag{16}$$
$$\tag{17}$$

We also have

$$P(\tilde{s}_{t+1}|H_t', a_t) = P(\tilde{s}_{t+1}|H_t', e_\phi(H_t'), a_t) \quad \text{(Because } \tilde{s}_{t+1} \perp\!\!\!\perp e_\phi(H_t)|H_t) \tag{18}$$
$$= P(\tilde{s}_{t+1}|e_\phi(H_t'), a_t) \quad \text{(By equation 11)} \tag{19}$$
$$= P(\tilde{s}_{t+1}|e_\phi(H_t), a_t) \quad \text{(Because } e_\phi(H_t') = e_\phi(H_t)) \tag{20}$$
$$= P(\tilde{s}_{t+1}|H_t, a_t), \tag{21}$$

which is exactly the condition for bisimulation specified by equation 13.

Similarly for reward, we have

$$P(r_t|H_t, a_t) = P(r_t|H_t, e_\phi(H_t), a_t) \quad \text{(Because } r_t \perp\!\!\!\perp e_\phi(H_t)|H_t) \tag{22}$$
$$= P(r_t|e_\phi(H_t), a_t) \quad \text{(By equation 12)} \tag{23}$$
$$\tag{24}$$

and

$$P(r_t|H'_t, a_t) = P(r_t|H'_t, e_\phi(H'_t), a_t) \quad \text{(Because } r_t \perp\!\!\!\perp e_\phi(H_t)|H_t) \tag{25}$$
$$= P(r_t|e_\phi(H'_t), a_t) \quad \text{(By equation 12)} \tag{26}$$
$$= P(r_t|e_\phi(H_t), a_t) \quad \text{(Because } e_\phi(H'_t) = e_\phi(H_t)) \tag{27}$$
$$= P(r_t|H_t, a_t), \tag{28}$$

which is exactly the condition for bisimulation specified by equation 14. Therefore, if $e_\phi$ satisfies equation 11 and equation 12, then the equivalence relation induced by $e_\phi$, $\sim_{e_\phi}$, is a bisimulation relation. $\square$

## B.2 Proof of Theorem 3.2

**Theorem 3.2** (Optimality of $\mathcal{L}$ yields bisimulation). *Suppose the model classes for the latent dynamics and reward distributions are sufficiently expressive. Then any global maximizer of the objective in equation 1 satisfies*

$$\hat{p}_\theta(\tilde{s}_t \,|\, \tilde{s}_{t-1}, a_{t-1}) = p(\tilde{s}_t \,|\, \tilde{s}_{<t}, o_{<t}, a_{<t}), \qquad \hat{p}_\theta(r_t \,|\, \tilde{s}_t, a_t) = p(r_t \,|\, o_{\leq t}, a_{\leq t}),$$

*for all $t$. These equalities imply the latent dynamics and reward sufficiency conditions, and therefore, by Lemma 3.1, the encoder-induced partition of histories is a bisimulation.*

*Proof.* By adding/subtracting the constant $\mathbb{E}[\log p(r_t \,|\, o_{\leq t}, a_{\leq t})]$, $\mathcal{L}$ can be re-written as

$$\mathcal{L} = \text{const} - \mathbb{E}\left[\sum_t D_{\text{KL}}(p(\tilde{s}_t \,|\, \tilde{s}_{<t}, o_{<t}, a_{<t}) \,\|\, \hat{p}_\theta(\tilde{s}_t \,|\, \tilde{s}_{t-1}, a_{t-1})) + D_{\text{KL}}(p(r_t \,|\, o_{\leq t}, a_t) \,\|\, \hat{p}_\theta(r_t \,|\, \tilde{s}_t, a_t))\right].$$

By nonnegativity of KL divergence, $\mathcal{L}$ is maximized if and only if both KL terms vanish almost surely, i.e.

$$\hat{p}_\theta(\tilde{s}_t \,|\, \tilde{s}_{t-1}, a_{t-1}) = p(\tilde{s}_t \,|\, \tilde{s}_{<t}, o_{<t}, a_{<t}), \qquad \hat{p}_\theta(r_t \,|\, \tilde{s}_t, a_t) = p(r_t \,|\, o_{\leq t}, a_t).$$

These identities assert that the history influences the true conditionals only through $(\tilde{s}_t, a_t)$, which is exactly the latent dynamics and reward sufficiency conditions. Lemma 3.1 then guarantees that the encoder-induced partition is a bisimulation. $\square$

## B.3 Proof that gradients with respect to the explicit occurrence of $\phi$ in the adversarial objective vanish

We follow a similar argument to that shown in Mescheder et al. (2017):

$$\frac{\partial}{\partial \phi} \mathbb{E} \left[ \sum_t \log \hat{p}_{\psi^*}(\tilde{s}_t | \tilde{s}_{<t}, o_{<t}, a_{<t}) \right] \tag{29}$$

$$= \frac{\partial}{\partial \phi} \mathbb{E} \left[ \sum_t \log p(\tilde{s}_t | \tilde{s}_{<t}, o_{<t}, a_{<t}, \phi) \right] \quad \text{(By equation 9)} \tag{30}$$

$$= \frac{\partial}{\partial \phi} \sum_t \mathbb{E}_{o_{<t}, a_{<t}, \tilde{s}_{<t}} \left[ \int p(o_t | o_{<t}, a < t) e_\phi(\tilde{s}_t | o_{\leq t}, a_{<t}) \log p(\tilde{s}_t | \tilde{s}_{<t}, o_{<t}, a_{<t}, \phi) do_t d\tilde{s}_t d \right] \tag{31}$$

$$= \frac{\partial}{\partial \phi} \sum_t \mathbb{E}_{o_{<t}, a_{<t}, \tilde{s}_{<t}} \left[ \int p(\tilde{s}_t | \tilde{s}_{<t}, o_{<t}, a_{<t}, \phi) p(o_t | \tilde{s}_{\leq t}, o_{<t}, a < t) \log p(\tilde{s}_t | \tilde{s}_{<t}, o_{<t}, a_{<t}, \phi) do_t d\tilde{s}_t \right] \tag{32}$$

$$= \frac{\partial}{\partial \phi} \sum_t \mathbb{E}_{o_{<t}, a_{<t}, \tilde{s}_{<t}} \left[ \int p(\tilde{s}_t | \tilde{s}_{<t}, o_{<t}, a_{<t}) \log p(\tilde{s}_t | \tilde{s}_{<t}, o_{<t}, a_{<t}, \phi) d\tilde{s}_t \right] \tag{33}$$

$$= \sum_t \mathbb{E}_{o_{<t}, a_{<t}, \tilde{s}_{<t}} \left[ \int p(\tilde{s}_t | \tilde{s}_{<t}, o_{<t}, a_{<t}, \phi) \frac{\partial}{\partial \phi} \log p(\tilde{s}_t | \tilde{s}_{<t}, o_{<t}, a_{<t}, \phi) d\tilde{s}_t \right] \tag{34}$$

$$= \sum_t \mathbb{E}_{o_{<t}, a_{<t}, \tilde{s}_{<t}} \left[ \int p(\tilde{s}_t | \tilde{s}_{<t}, o_{<t}, a_{<t}, \phi) \frac{\nabla_\phi p(\tilde{s}_t | \tilde{s}_{<t}, o_{<t}, a_{<t}, \phi)}{p(\tilde{s}_t | \tilde{s}_{<t}, o_{<t}, a_{<t}, \phi)} d\tilde{s}_t \right] \tag{35}$$

$$= \sum_t \mathbb{E}_{o_{<t}, a_{<t}, \tilde{s}_{<t}} \left[ \nabla_\phi \int p(\tilde{s}_t | \tilde{s}_{<t}, o_{<t}, a_{<t}, \phi) d\tilde{s}_t d \right] \tag{36}$$

$$= \sum_t \mathbb{E}_{o_{<t}, a_{<t}, \tilde{s}_{<t}} \left[ (\nabla_\phi 1) \right] \tag{37}$$

$$= 0. \tag{38}$$

## C  MODEL ARCHITECTURE

Our model consists of four main components: an observation encoder, a latent dynamics model, a reward model, and an adversary network.

**Observation Encoder.**  The encoder maps the agent's history up to time $t$, $(o_{\leq t}, a_{<t})$, into a distribution over the latent state $\tilde{s}_t$. The architecture is composed of a convolutional neural network (CNN) followed by a GRU. Unlike Dreamer, which shares a GRU between the encoder and dynamics model, we use a dedicated GRU for the encoder. The encoder outputs a mean vector $\mu_t$, normalized to have $\ell_2$ norm equal to one. We then sample $\tilde{s}_t$ from a Gaussian distribution with mean $\mu_t$ and fixed variance 0.01. This normalization and variance-fixing introduces an information bottleneck: only a fixed amount of information can flow through the latent state, encouraging abstraction.

**Latent Dynamics Model.**  The dynamics model resembles that of Dreamer, with both stochastic and deterministic components. It maintains a deterministic hidden state $h_t$ via a GRU and outputs the parameters of a Gaussian distribution over $\tilde{s}_t$. The mean is again normalized to unit norm, while the variance is learned but constrained to be isotropic (identical across latent dimensions).

**Reward Model.**  Our reward model differs slightly from Dreamer's: it conditions only on the stochastic latent state $\tilde{s}_t$ (and the current action $a_t$), not on the deterministic hidden state $h_t$. It outputs the parameters of a Gaussian distribution over rewards $r_t$.

**Adversary Network.**  Finally, the adversary network shares a similar architecture with the encoder (CNN + GRU). The adversary additionally conditions on $s_t$ through an auxiliary input before the GRU, and predicts the *next* latent state $\tilde{s}_{t+1}$ instead of the current one. Similar to the dynamics model, the adversary network outputs a unit-norm mean and an isotropic variance vector.

---

**Algorithm 1:** VIBES Model Training Algorithm

---

**Inputs:** Replay buffer $\mathcal{D}$ with sequences $(o_{0:T}, a_{0:T-1}, r_{0:T-1})$
**Models:** Encoder $e_\phi: (o_{\leq t}, a_{<t}) \mapsto \tilde{s}_t \sim \mathcal{N}(\mu_t, \sigma_e^2 I)$ with $\|\mu_t\|_2 = 1$ and fixed $\sigma_e^2$;
    Deterministic state update $h_t = f_\theta(h_{t-1}, \tilde{s}_{t-1}, a_{t-1})$ (GRU);
    Dynamics $p_\theta(\tilde{s}_t \mid h_{t-1}, \tilde{s}_{t-1}, a_{t-1})$: Gaussian with unit-norm mean, isotropic variance;
    Reward $p_\theta(r_t \mid \tilde{s}_t, a_t)$: Gaussian;
    Adversary $\hat{p}_\psi(\tilde{s}_t \mid \tilde{s}_{<t}, o_{<t}, a_{<t})$: Gaussian with unit-norm mean, isotropic variance
**Hyperparams:** Learning rates $\eta_\theta, \eta_\phi, \eta_\psi$ with $\eta_\psi > \eta_\phi$; batch size $B$; sequence length $T$
**Initialize:** Randomly initialize $\phi, \theta, \psi$; set $h_0 = \mathbf{0}$

1 **for** *training steps* $= 1, 2, \ldots$ **do**
2    Sample $B$ trajectory chunks of length $T$ from $\mathcal{D}$
    // Encode latents with reparameterization
3    **for** $t = 0$ **to** $T$ **do**
4       $(\mu_t, \sigma_e^2) \leftarrow e_\phi(o_{\leq t}, a_{<t})$, enforce $\|\mu_t\|_2 \leftarrow 1$
5       Sample $\epsilon_t \sim \mathcal{N}(0, I)$,   $\tilde{s}_t \leftarrow \mu_t + \sigma_e \odot \epsilon_t$
    // Update deterministic hidden state, then compute latent
       dynamics and reward
6    **for** $t = 1$ **to** $T$ **do**
7       $h_t \leftarrow f_\theta(h_{t-1}, \tilde{s}_{t-1}, a_{t-1})$              // GRU update
8       Compute $p_\theta(\tilde{s}_t \mid h_{t-1}, \tilde{s}_{t-1}, a_{t-1})$     // latent transition
9       Compute $p_\theta(r_t \mid \tilde{s}_t, a_t)$              // reward model
    // Adversary predictions (condition on full history)
10   Compute $\hat{p}_\psi(\tilde{s}_t \mid \tilde{s}_{<t}, o_{<t}, a_{<t})$ for $t = 1{:}T$
    // Per-dimension non-positivity clipping of the ratio term
11   **for** $t = 1$ **to** $T$ **do**
12       **for** *each latent dim* $i$ **do**
13          $\ell_t^{(i)} \leftarrow \sum_{b=1}^{B} \log \dfrac{p_\theta(\tilde{s}_{t,b}^{(i)} \mid h_{t-1,b}, \tilde{s}_{t-1,b}, a_{t-1,b})}{\hat{p}_\psi(\tilde{s}_{t,b}^{(i)} \mid \tilde{s}_{<t,b}, o_{<t,b}, a_{<t,b})}$
14          $\tilde{\ell}_t^{(i)} \leftarrow \min(\ell_t^{(i)}, 0)$       // clip so term is non-positive
    // Objective (minibatch average)
15   $\mathcal{L} \leftarrow \dfrac{1}{TB} \sum_{t=1}^{T} \left[ \sum_i \tilde{\ell}_t^{(i)} + \sum_{b=1}^{B} \log p_\theta(r_{t,b} \mid \tilde{s}_{t,b}, a_{t,b}) \right]$
    // Outer ascent updates (treat $\psi^*$ as fixed; Appendix B.3)
16   $\theta \leftarrow \theta + \eta_\theta \nabla_\theta \mathcal{L}; \quad \phi \leftarrow \phi + \eta_\phi \nabla_\phi \mathcal{L}$
    // Inner ascent update on adversary (maximize equation 9)
17   $\mathcal{J}_\psi \leftarrow \dfrac{1}{TB} \sum_{b=1}^{B} \sum_{t=1}^{T} \log \hat{p}_\psi(\tilde{s}_{t,b} \mid \tilde{s}_{<t,b}, o_{<t,b}, a_{<t,b})$
18   $\psi \leftarrow \psi + \eta_\psi \nabla_\psi \mathcal{J}_\psi$

---

## C.1 MODEL TRAINING ALGORITHM

## D LATENT SPACE ANALYSIS

## E LLM USAGE

During the preparation of this manuscript, we used a large language model (LLM) to assist with polishing the exposition in select sections. All technical content, derivations, algorithm design, and experimental results were authored and verified entirely by the human authors. The LLM's role was limited to improving clarity, conciseness, and readability of the text, with final edits and approval performed by the authors.

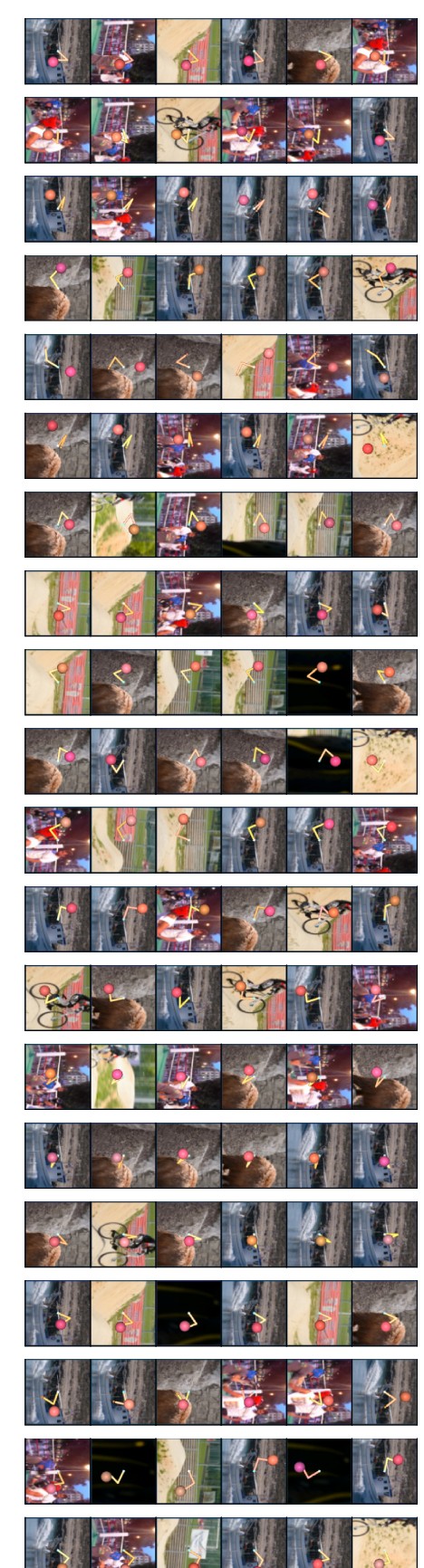

Figure 5: **Nearest-neighbor analysis in latent space (all 20 queries).** For each of 20 query observations (left-most image in each row), we display the five nearest neighbors in the learned latent space beneath it (standardized $L_2$ distance). The visualization shows invariance to visual distractors and clustering of rotationally symmetric arm–goal configurations.

