# OpenReview forum: "Distractor-Robust Reinforcement Learning via Variational Bisimulation"
_ICLR.cc/2026/Conference — Submitted to ICLR 2026_

### Official Review · Reviewer_tGnB · 2025-10-24

**Soundness:** 2
**Presentation:** 2
**Contribution:** 2
**Rating:** 4
**Confidence:** 3

**Summary:**

This paper presents a novel objective function for model learning in model-based reinforcement learning.
Specifically, it proposes a modification of the objective function used in Dreamer (Hafner et al., 2020).
In Dreamer, the observation $o_t$ at time $t$ is assumed to depend solely on the latent state $s_t$.

In contrast, this paper removes such an assumption and instead learns the distribution of the latent state $s_t$ conditioned on the past observations $o_{<t}$ and actions $a_{<t}$.
In the resulting objective function, the reconstruction loss is omitted, and a density ratio between  $ p ( \hat{s}_{t} \mid \hat{s}_{t-1}, \hat{a}_{t-1})$ and $p( \hat{s}_t \mid \hat{s}_{< t}, \hat{a}_{< t})$ is introduced.

The proposed method is evaluated on continuous control tasks with visual distractions, where it outperforms baseline methods in the reported experiments.

**Strengths:**

- The proposed modification to the model structure and the objective function is interesting and appears to be novel.

- The advantages of the proposed method are clearly demonstrated in the experiments.

The proposed modification of the model structure and objective function seems reasonable.
Since more recent variants of Dreamer, such as DreamerV2 and DreamerV3, still employ architectures similar to that of the original Dreamer, the proposed method may also improve the performance of these newer variants.

**Weaknesses:**

- The presentation of the proposed method could be improved.

- The experiments lack comparison with several recent and well-known methods.

- A more extensive discussion on the relation to recent methods is desirable.

In particular, the adversarial network component of the proposed method is insufficiently explained.
Although the derivation becomes clearer after reading the appendix, I believe that Equations (6)–(9) should be incorporated into the main text to provide intuition for why the model $\hat{p} ( \tilde{s}_{t} | \tilde{s}_{<t}, o_{<t}, a_{<t})$ is referred to as the “adversary network.”

While the experiments show that the proposed method outperforms Dreamer, the baseline methods used appear to be somewhat outdated.
More recent baseline methods include:

[R1] TD-MPC2: Scalable, Robust World Models for Continuous Control. Nicklas Hansen, Hao Su, Xiaolong Wang. ICLR 2024.
[R2] Towards General-Purpose Model-Free Reinforcement Learning. Scott Fujimoto, Pierluca D'Oro, Amy Zhang, Yuandong Tian, Michael Rabbat. ICLR 2025.

Although a variant of DrQ is used in the comparison, I would recommend using DrQv2 [R3]

[R3] Mastering Visual Continuous Control: Improved Data-Augmented Reinforcement Learning. Denis Yarats, Rob Fergus, Alessandro Lazaric, Lerrel Pinto. ICLR 2022.

Recent methods that leverage model learning, such as TD-MPC2 and Mr.Q, also remove the reconstruction loss.
Therefore, removing the reconstruction loss itself is not a novel idea, and I recommend adding a discussion clarifying how the proposed approach relates to these recent methods.

In addition, newer variants of Dreamer, such as DreamerV2, DreamerV3, and Dreamer4, still employ reconstruction losses, and thus may benefit from the proposed objective function.
It would strengthen the paper to include experiments demonstrating the effect of the proposed objective function on these more recent Dreamer variants.

*Minor Comment*

In the appendix, line 538, the text reads: “Similar to our derivation of the Dreamer model objective.”
However, it seems that the authors are actually referring to the derivation of the objective function in PlaNet (Hafner et al., 2019).
Although Dreamer builds upon the PlaNet model, it would be more accurate to explicitly cite the reference on which the derivation is based.

**Questions:**

- Please elaborate on how the proposed method relates to recent approaches that omit the reconstruction loss.

- It would also be valuable to include comparisons with more recent baseline methods and to evaluate the proposed objective function when integrated into a newer variant of Dreamer.

---

> ### Author Response · Authors · 2025-12-01
> **Response to Reviewer tGnB**
>
> Thank you for your constructive feedback.
>
> To address the weakness regarding missing baselines, we have run one additional method, Deep Bisimulation for Control (DBC) [1], and hope to complete more baselines before the end of the discussion phase. DBC underperformed relative to VIBES, as shown in the table below:
>
> |     | Ball in cup     | Cartpole         | Cheetah         | Finger-Spin    | Reacher         | Walker         |
> |-----|-----------------|------------------|-----------------|----------------|-----------------|----------------|
> | DBC | $98.9 \pm 44.8$ | $118.5 \pm 20.8$ | $61.7 \pm 49.0$ | $10.2 \pm 6.7$ | $89.0 \pm 21.4$ | $29.3 \pm 2.2$ |
>
>
> Regarding the clarity of our method’s presentation, we agree that Equations 6–9 should appear in the main text. We initially omitted them due to space constraints, but they are indeed important for understanding the objective. We have now moved these equations into the main body of the paper.
>
> On the question of why the model $\hat{p} ( \tilde{s}{t} | \tilde{s}{<t}, o_{<t}, a_{<t})$ is referred to as the adversary network: this component is trained with an objective that is the opposite of the objective used for the main model. As a result, the adversary network and the main model participate in an adversarial game. This terminology also connects our approach to prior work in variational inference that employs similar adversarial structures, such as Adversarial Variational Bayes [2], although the correspondence between our “adversary” and their “discriminator” is not one-to-one.
>
> More broadly, summarizing the relationship between our work and prior state-abstraction methods is challenging because the literature is large and diverse. Many existing approaches rely on hand-designed data augmentation strategies, which our method avoids entirely. Unlike most prior approaches, we also provide theoretical guarantees about the type of state representation the method learns. Compared to other bisimulation-based approaches—which typically optimize a continuous relaxation of bisimulation in the form of bisimulation metrics [1, 3]—our method learns bisimulation relations directly and integrates naturally with variational-inference-based world models such as Dreamer.
>
> Finally, you are correct that the derivation we reference originates in the PlaNet paper (Hafner et al., 2019), and we have corrected this citation in the revised manuscript.
>
> References
>
> [1] Zhang, Amy, et al. Learning Invariant Representations for Reinforcement Learning without Reconstruction. ICLR.
>
> [2] Mescheder, Lars, Sebastian Nowozin, and Andreas Geiger. Adversarial Variational Bayes: Unifying Variational Autoencoders and Generative Adversarial Networks. ICML, 2017.
>
> [3] Castro, Pablo Samuel. Scalable Methods for Computing State Similarity in Deterministic Markov Decision Processes. AAAI, 2020.

---

### Official Review · Reviewer_P65f · 2025-10-29

**Soundness:** 3
**Presentation:** 3
**Contribution:** 3
**Rating:** 4
**Confidence:** 3

**Summary:**

This work proposes to use a bistimulation-driven abstraction in order to learn useful latent representations to tackle the visual distractors  problem with model-based online RL. This work argues that that previous works offer small benefits when learning the full-pixel reconstruction for tasks where visual distractors exist, thus, by replacing the pixel-reconstruction term of a model-based method, Dreamer, with an adversarial objective that enforces a bistimulation-based abstraction, more task-related latent representation can be learned.
The authors experimented on the DeepMind Distracting Control Suite, and empirically showed improved performance on various tasks.

**Strengths:**

- The paper is clearly written and easy to follow.
- Although as written by the authors, learning state abstractions to tackle the visual distractor problem is not novel, the proposed method of using bisimulation is well motivated and seems intuitive and novel. Although enforcing bisimulation was previously used [1][2], to my understanding the novelty lies on the way the authors use it, i.e. derive it as an adversarial objective rather than the original balancing term in [1][2], is quite novel to my knowledge.
- The results on a designed high-dimensional problem are important and show the efficiency of the method.

**Weaknesses:**

- Some important model-based related works, using other types of state abstractions, that also specifically tackle the visual distractor problem in continuous control might be missing during empirical comparison [2][3]. [2] is a directly related bisimulation work with different formulation.
- Additionally, not really a weakness but, I’m not really sure why this work is applied to the first iteration of dreamer. It would be great if further performance gain can be shown if this work were built with Dreamerv3 [4].
- Also not really a weakness but, instead of presenting the main results as a table, it would be nice if the training timesteps/return plot for tasks in table 1 is included (i.e. same as figure 3).
- Hyperparameters are not clearly listed

- Minor issues such as:
  - missing texts at the start of row 90
  - Typo in row 123: relaitons -> relations
  - The abbreviation PGM, is not explained in figure 1.a and 1.b

**Questions:**

I will repost some suggestions/questions that were mentioned in the weakness section here for clarity.

$$\textbf{Suggestions}$$
S1:Presentation-wise: fix the typos, clearly state the hyperparameters
S2: Presentation-wise: Add the training timesteps/return plot for table 1
S3: Presentation-wise: very minor, but somehow rendering page 8 is very heavy (I’ve tested on several devices across different platforms), I suspect it has something to do with the Figure 4a. It would be great if the authors can check.
S4: I would prefer to see some ablation study on the *type* of distractions, i.e. using only one type of distraction when evaluating. This could help readers understand which type of distractions bisimulation is strong against and weak against. (c.f. something like Section 8.2 in this paper https://openreview.net/forum?id=dce6ZGkJ1Z)

$$\textbf{Questions}$$
Q1: Is there a particular reason to use Dreamerv1 as the base algorithm? It has been 3-4 years.
Q2: I’m a little bit confused about the adversarial term, which is the most important term to my understanding that pushes for dynamics sufficiency.  What do the authors think about the empirical results where for some tasks (ball-in-cup catch, finger-spin) the adversarial term is quite important, but for some tasks (carpole-swingup, cheetah-run, reacher-easy, walker-walk) adding it only achieve trivial gains?
Q3: For the UMAP analysis, following Q2, since the final performance is similar with or without the adversarial term for reacher-easy, I wonder how the UMAP analysis looks like without the adversarial term? If possible, it would be good to see the same UMAP analysis for tasks where significant performance difference were achieved with or without the adversarial term.
Q4: How efficient is the proposed formulation to adopt adversarial bisimulation objectives compared to other bisimulation-principled formulations? Direct comparison with other bistimulation-based work [2] should be included. For [2], their code is public available at https://github.com/bit1029public/HRSSM

---

[1] P. S. Castro, Scalable methods for computing state similarity in deterministic Markov Decision Processes, AAAI 2020
[2] R. Sun et al., Learning Latent Dynamic Robust Representations for World Models, ICML 2024
[3] C. Zhu et al., RePo: Resilient Model-Based Reinforcement Learning by Regularizing Posterior Predictability, NeurIPS 2023
[4] D. Hafner et al., Mastering diverse control tasks through world models, Nature 2025

---

> ### Author Response · Authors · 2025-11-20
> **Response to reviewer P65f**
>
> Thank you for the constructive and insightful feedback. We appreciate the opportunity to clarify and improve the manuscript.  We will upload an updated version shortly.
> # Weaknesses
> - Thank you for pointing out these oversights. We have run [2] and are working to run [3] before the end of the discussion period, along with additional baselines suggested by the other reviewers. The corresponding results are in the table below.
>
> |     | Ball in cup     | Cartpole         | Cheetah         | Finger-Spin    | Reacher         | Walker         |
> |-----|-----------------|------------------|-----------------|----------------|-----------------|----------------|
> | DBC | $98.9 \pm 44.8$ | $118.5 \pm 20.8$ | $61.7 \pm 49.0$ | $10.2 \pm 6.7$ | $89.0 \pm 21.4$ | $29.3 \pm 2.2$ |
>
> - We did not build on Dreamer v1 specifically; rather, we built on the underlying variational-inference formulation shared by Dreamer v1, v2, and v3—that is, learning a latent state representation along with dynamics and reward models, and then planning or acting in the latent space. Our architecture, however, differs substantially from all Dreamer variants. For instance, our latent state is neither the continuous vector of Dreamer v1 nor the discrete code of Dreamer v3. Instead, we use a unit-sphere–constrained latent with fixed variance, which provides an explicit information bottleneck and limits the amount of information that the model can encode.  It would indeed be interesting to investigate whether incorporating architectural advancements from Dreamer v2/v3—such as symlog regression for reward prediction or distributional critics—could further improve our method. We chose not to include these modifications to avoid confounding the contribution of our variational modeling objective.
> - We will add reward curves, clarified hyperparameters, and fixed the noted typos in the updated manuscript.
> # Suggestions / Questions
> - S3: We will address this rendering issue in the revised manuscript.
> - S4: Thank you for the suggestion—we will look into running this ablation.
> - Q1: (See response under Weakness 2.)
> - Q2: Thank you for the insightful question. As we note briefly in the ablation studies subsection of Sec. 4.1, the adversarial term can become unnecessary in environments where the underlying MDP state is fully observable and transitions are deterministic. In such cases, the optimal adversary can already “perfectly” predict the next state from the history (modulo encoder noise), causing the adversarial term to reduce to a constant and leaving the bisimulation objective effectively unchanged. Empirically, we observed that the tasks where the adversary contributed little were those in which distractors obscured the true state the least. We are currently exploring additional analyses to validate this explanation.
> - Q3 & Q4: Both are helpful suggestions. We will explore running these experiments for the revised version.

---

### Official Review · Reviewer_PfgH · 2025-10-30

**Soundness:** 3
**Presentation:** 2
**Contribution:** 3
**Rating:** 6
**Confidence:** 4

**Summary:**

This paper introduces a Model-Based Reinforcement Learning method called VIBES, designed to address the limitations of traditional reconstruction-based objectives that waste representational capacity on reconstructing task-irrelevant distractors. VIBES replaces pixel-level reconstruction with an adversarial variational objective, constrained by reward sufficiency and dynamics sufficiency, ensuring that the learned latent states retain only task-relevant information. The authors theoretically prove that, under mild assumptions, the global optimum of this objective corresponds to an encoder that induces bisimulation relations, thus preserving control-relevant structure while discarding irrelevant variations. Empirically, VIBES can directly replace the model-learning component in Dreamer, achieving state-of-the-art performance on the Distracting Control Suite (DCS) and demonstrating strong results on high-dimensional vector-state control tasks such as a 100-link Swimmer.

**Strengths:**

1. The proposed VIBES framework is concise and integrates seamlessly into existing MBRL pipelines such as Dreamer. By simply replacing the pixel reconstruction term with an adversarial variational objective, VIBES maintains the overall architecture while significantly improving representation learning efficiency.

2. Experiments on the Distracting Control Suite (DCS) show that VIBES achieves large performance gains over visual RL baselines such as SAC+RAD and CURL, demonstrating superior robustness and a strong ability to focus on task-relevant information in visually complex environments.

**Weaknesses:**

**1. Dependence on the quality of the Adversary Network**

The quality of the learned representation heavily depends on the performance of the Adversary Network. In practice, the stability and effectiveness of VIBES can be sensitive to the adversary's training quality. Poor adversary learning may weaken both theoretical guarantees and overall performance. Quantitatively evaluating the performance of the Adversary Network would make the paper's claims more convincing.

**2. Lack of comparison with recent bisimulation-based methods**

The paper does not include direct comparisons with bisimulation-based methods. Although these works are cited, there is a lack of direct performance or learning curve comparisons, e.g., with DBC[1], MICO[2], and approaches that apply bisimulation metrics within world models[3], making it difficult to quantitatively assess the advantages of the proposed method. In addition, the paper does not present learning curve plots for its experiments; including them would help clarify the convergence behavior more clearly.

**3. Limited experimental generality**

The empirical evaluation focuses mainly on the Distracting Control Suite (DCS) and the extended Swimmer environment. While the results are impressive, the study would be more convincing if additional experiments on standard DeepMind Control (DMC) tasks or other benchmarks were included to demonstrate broader applicability.

[1] Learning Invariant Representations for Reinforcement Learning without Reconstruction. In International Conference on Learning Representations.
[2] MICo: Improved representations via sampling-based state similarity for Markov decision processes. Advances in Neural Information Processing Systems.
[3] Learning Latent Dynamic Robust Representations for World Models. In International Conference on Machine Learning.

**Questions:**

1. Could the authors provide a quantitative evaluation of the Adversary Network's performance to better validate its role in representation learning?

2. Could the authors include direct comparisons with bisimulation metric learning methods and add experiments on standard DeepMind Control (DMC) benchmarks to more comprehensively demonstrate the advantages and generalization ability of the proposed approach?

---

> ### Author Response · Authors · 2025-12-01
> **Response to Reviewer PfgH**
>
> Thank you for the insightful feedback.  Below are responses to each of the concerns you raised.
>
> 1. We generally found that adversary training was fairly stable and one set of adversary hyperparameters was suitable for all tasks.  To better make this point, we will provide quantitative metrics of adversary network performance in the updated manuscript.  We observed that for most environments, the adversary was slightly better at predicting the next state compared to the dynamics network, which is expected given that the adversary has more information available with which to make predictions.
>
> 2.  Thank you for pointing out these highly relevant works.  So far, we have run DBC, and we hope to run additional baselines by the end of the discussion period.  As can be seen by the table below, DBC underperforms compared to VIBES:
>
> |     | Ball in cup     | Cartpole         | Cheetah         | Finger-Spin    | Reacher         | Walker         |
> |-----|-----------------|------------------|-----------------|----------------|-----------------|----------------|
> | DBC | $98.9 \pm 44.8$ | $118.5 \pm 20.8$ | $61.7 \pm 49.0$ | $10.2 \pm 6.7$ | $89.0 \pm 21.4$ | $29.3 \pm 2.2$ |
>
> 3.  Thank you for this suggestion.  We will work to include experiments on DMC in the final version of the manuscript.

---

### Official Review · Reviewer_u3AN · 2025-10-31

**Soundness:** 2
**Presentation:** 1
**Contribution:** 2
**Rating:** 2
**Confidence:** 4

**Summary:**

This paper proposes Variational Inference for Bismulation-based Encoded States (VIBES), a model-based RL method that learns a recurrent latent dynamics model on bismulation representations. Unlike reconstruction-based approaches, which try to capture everything, the bisimulation representation is invariant to task-irrelevant distractors and only captures what's necessary to predict the dynamics and rewards. The main contribution is a new objective to learn this representation by regularizing the latent dynamics against a separate history encoder, which is mathematically shown to induce a bismulation representation. The method is evaluated on the distracted DM control suite, where it outperforms a suite of RL baselines.

**Strengths:**

1. This paper addresses an important problem in model-based reinforcement learning: image-based dynamics models are typically trained with pixel reconstruction, and therefore waste model capacity by encoding task-irrelevant details. The paper proposes a reasonable solution grounded in the literature of bisumulation metrics.
2. The paper conducts detailed ablation studies of the proposed method, justifying the contribution of each objective term.

**Weaknesses:**

1. The mathematical derivation is not sound. Specifically, assumption (2) in Lemma 3.1 leads to a chicken-and-egg problem. The argument that the authors make is: assume latent $s_t$ is independent of the history $H_t$ conditioned on the previous latent $s_{t-1}$ and action $a_{t-1}$, then the history encoder's embedding establishes an bisimulation relation on the space of histories. However, if $s_t$ is independent of history conditioned on $s_{t-1}$ and $a_{t-1}$, then it must have already captured whatever is necessary to predict the future latent state. This is exactly the goal of bisimulation. With this assumption, it becomes a tautology that the history encoder induces a bisumulation. For this reason, it is unclear if the proposed method actually learns a bisimulation.
2. The probablistic graphical model in Figure 2 lack justification. See Questions 1 for details.
3. The experiments lack comparsions to relevant model-based RL baselines (reconstruction-based and non-reconstruction-based) and bimulation baselines [1, 2, 3, 4, 5, 6]. The problem of task-irrelevant distractions has been studied by a rich line of work, non of which is referenced or compared to. This is a serious omission and is the main reason I recommend a rejection.

**Questions:**

**Major**
1. Can you explain the intuition behind the PGM in Figure 1 in more detail? I struggle to understand it, because if $o_t$ sees all $o_{<t}$ then there's no need for $s$.
2. Can you address the chicken-and-egg problem mentioned in Weakness 1?
3. How does equation 9 use Gibbs inequality?
4. Since the adversary network is also a GRU, it intrinsically contains a Markovian state. Isn't this a duplicate of the main model?
5. In line 595, you mention that the dynamics term is clipped to be non-positive. What happens if you remove the clipping? Can you ablate this design choice?

**Minor**

6. line 90: sentence doesn't have a subject

References:

[1] Danijar Hafner, Timothy Lillicrap, Jimmy Ba, Mohammad Norouzi. Dream to Control: Learning Behaviors by Latent Imagination. ICLR 2020.

[2] Nicklas Hansen,  Xiaolong Wang,  Hao Su. Temporal Difference Learning for Model Predictive Control. NeurIPS 2022.

[3] Xiang Fu, Ge Yang, Pulkit Agrawal, Tommi Jaakkola. Learning Task Informed Abstractions. NeurIPS 2021.

[4] Chuning Zhu, Max Simchowitz, Siri Gadipudi, Abhishek Gupta. RePo: Resilient Model-Based Reinforcement Learning by Regularizing Posterior Predictability. NeurIPS 2023.

[5] Tongzhou Wang, Simon S. Du, Antonio Torralba, Phillip Isola, Amy Zhang, Yuandong Tian. Denoised MDPs: Learning World Models Better Than the World Itself. NeurIPS 2022.

[6] Amy Zhang, Rowan McAllister, Roberto Calandra, Yarin Gal, Sergey Levine. Learning Invariant Representations for Reinforcement Learning without Reconstruction. ICLR 2021.

---

> ### Author Response · Authors · 2025-11-16
> **Responose to Reviewer u3AN**
>
> Thank you for your thoughtful and constructive review. We’d like to clarify a few points brought up in your review.
>
> **Weaknesses:**
> 1) Thank you for raising this point. Our intention was not to assume conditions (2) and (3) (dynamics and reward sufficiency), but rather to prove that they hold when the model optimizes our objective $\mathcal{L}$. The logical structure was unfortunately obscured by the order in which the results were presented.
> To clarify:
>     - Lemma 3.1: If conditions (2) and (3) hold, then $\tilde{s}$ induces a bisimulation relation.
>     - Theorem 3.2: Any global optimum of $\mathcal{L}$ necessarily satisfies conditions (2) and (3).
>
> We are restructuring Section 3.2 to make this logical flow explicit in the following way: Lemma 3.1 will now establish that the global optimum of $\mathcal{L}$ yields a state representation that satisfies (2) and (3) (dynamics and reward sufficiency). The proof is the same as was given for our current theorem 3.2 in B.2 of the appendix. Lemma 3.2 will establish that (2) and (3) imply that $\tilde{s}$ encodes a bisimulation relation.  This is the same as our current lemma 3.1 in the manuscript, and the proof is in B.1 of the appendix.
> Theorem 3.3 will simply chain lemmas 3.1 and 3.2, to prove that global optima of $\mathcal{L}$ yield  bisimulation relations.
>
> 2. (see Q1)
>
> 3. Thank you for pointing out these relevant related works.  We will run comparisons to as many of these methods as possible and post the results once completed.
>
> **Questions**
> 1. We chose the PGM structure specifically to:
>     1. **Relax Dreamer's conditional independence assumption** that $o_t \perp o_{< t} | s_t $, which forces $s_t$ to encode *all* past information necessary for predicting $o_t$, including distractors.
>     2. **Retain bisimulation-relevant constraints**, i.e., $s_t$ must still predict $r_t$ and $s_{t+1}$ without access to past observations.
> It is true that $o_t$ could, in principle, extract all predictive information from $o_{< t}$ alone.  However, because $r_t$ and $s_{t+1}$ are modeled to depend only on $s_t$ and $a_t$, $s_t$ is still forced to retain all task-relevant information. This is precisely the reward/dynamics sufficiency required for bisimulation.  We will clarify this motivation in the revision.
> 2. See response to Weakness 1.
> 3. We now include the full derivation in the text. In short:
> Rewriting the expectation and applying Gibbs inequality gives:
> $E_p [\log p] \ge E_p [\log \hat{p}] $
> with equality only if $p=\hat{p}$.  This establishes Eq. 9 directly. The details will be added to the manuscript.
> 4. The adversary does maintain its own recurrent hidden state, but it is not forced through the same information bottleneck. Its purpose is to detect any information about the future latent state that is not present in the dynamics model state. This makes its role distinct from the GRU in the main model. It is an interesting area of future work to investigate whether representations can be shared between the adversary and main model.
> 5. We have run initial tests without clipping and observed less stable learning. Occasionally, the adversary lags behind the main model, thus providing a biased gradient signal for certain latent state components, which clipping helps prevent. We will include full ablations in the revised version.
> 6. Thank you for pointing out this typo.  We meant to say “Variational inference (VI) provides a general framework for learning latent-variable models.”  We have corrected it in the manuscript.

---

### Official Review · Reviewer_EoRX · 2025-11-08

**Soundness:** 3
**Presentation:** 4
**Contribution:** 4
**Rating:** 8
**Confidence:** 3

**Summary:**

A primary motivation of the paper is to tackle the issue of distractors in Model-base Reinforcement Learning (MBRL), where the model training objective is not matched to the control problem. This leads to the model learning features of the data that is unnecessary to the control problem. The paper proposes the learning of state representations that satisfy the bisimulation relations that encode the information relevant to the control problem (i.e, the distribution over the next state and the reward). To this, the paper proposes a variational objective that they show captures the bisimulation relations.

**Strengths:**

- Great introduction setting up a clear motivation.
- Writing is great and often clear. Figure captions clearly state the key takeaway which is nice.
- Experimental results look promising.
- Good ablation studies. I have suggested some more in the Questions section (Point 4).
- Discussion sections are thorough, with a good limitations section.
- I appreciate the detailed derivations in the Appendix.

**Weaknesses:**

- Please provide references at the following statements:
	- Line 54: "... domain-specific strategies, such as image augmentations, ..."
	- Line 45: "This problem has been shown to significantly degrade the performance of otherwise strong algorithms such as Dreamer (Hafner et al., 2020), which performs well on clean image-based benchmarks but struggles in the presence of distractors" — is there a reference for where this study has been carried out? I.e., the authors say that "this problem has been *shown*..." — where?
- Line 142: "...we can equivalently think of bisimulation relations as partitioning histories." — could this statement be expressed mathematically for clarity? Particularly since this is the view that is used in the paper.
- Appendix C does not detail sufficient information for reproducibility. In particular, it requires the reader to go to other papers (which aren't directly cited in the specific sentences) to find specific details. It would benefit the paper to have the full specification of the architecture (such that the experiments are fully reproducible) in the Appendix.
### Minor Suggestions
- Line 128-130 (bisimulation relation) — please add "and" after the first condition so it is clear that this is a logical conjunction.
- Some notation has incorrect underscoring. E.g.,
	- Figure 2 caption: $\tilde{s}\_{t} + 1$ should be $\tilde{s}_{t+1}$ (I think).
	- Line 317: ... $o<t$... should be $o_{<t}$ (I think).
- Appendix D looks empty. I am guessing that Figure 5 is supposed to be there? Perhaps a description could be added to Appendix D (this should be done to help the reader) — this will also push Appendix E to be after the image.
- Please mention in the main text that Appendix C contains the architectural setup for reproducing the experiments.

**Questions:**

- Line 55: "... whose use defeats the purpose of RL in the first place." — why is this the case?
- Section 2.1: Is the section title used as the start of the sentence?
- Figure 1: Shouldn't the actions $a_i$ depend on the observations $o_i$?
- Final paragraph of Section 2.1: Is there an ablation study you could carry out to show that allowing the observations to affect future observations "removes the pressure for $s_t$ to transmit irrelevant predictive information"? E.g., what would the experiment results look like if everything in your method is kept the same, but the graphical model of Figure 1a is used.
- Line 186: How was the fixed variance chosen? What would the results look like if this was changed?

---

> ### Author Response · Authors · 2025-12-01
> **Response to Reviewer EoRX**
>
> Thank you for your constructive feedback. We've addressed each of your concerns below.
>
> # Weaknesses
>
> 1a. Reinforcement learning approaches that use domain-specific data augmentations include Curl [1], DrQ [2], DrQ-v2 [3], RAD [5], PSE [6], and Network Randomization [7].  We will add these citations to the manuscript.
>
> 1b. Dreamer was shown to perform poorly on environments with distractors in [8] and [9], while Dreamer-like algorithms were shown to perform poorly with distractors in [10] and [11]. We will add these citations to the manuscript.
>
> 2. Yes we can make this mathematically precise.  As we mention in Lemma 3.1, in the case of partially observable environments, we can think of a bisimulation relation as partitioning state-observation *histories*, *i.e.,* let $H_t = (o_0, a_0,..., o_{t-1}, a_{t-1}, o_t)$ be the history up to time $t$.  Let $ s_t = e_{\phi}(H_t)$ be the abstract state produced by an encoder $e_{\phi}$.  We say that the encoder induces an equivalence relation $\sim_{e_{\phi}}$ on histories, where $ H_t \sim_{e_{\phi}} H_{t}^{'} $ means that  we map $ H_t $ and $H_{t}^{'} $ to the same abstract state, *i.e.*, $e_{\phi}(H_t) = e_{\phi}(H_{t}^{'})$.  The relation $  \sim_{e_{\phi}} $ is a bisimulation over histories if conditions (11) and (12) are met.  We will clarify this for the final draft.
>
> 3.  We will include the full details of the architecture in the final version of the manuscript.
>
> # Questions
>
> 1. Data augmentation typically require domain-specific knowledge, *i.e.*, what sorts of augmentations work well for that domain.  This partially defeats the purpose of RL, whose goal is to solve tasks with no domain-specific knowledge.
>
> 2.  No, the first sentence in sec. 2.1 is simply "Variational inference (VI) provides a general framework for learning latent-variable models."
>
> 3. Yes, if we assume the existence of a feedback policy $\pi(a_i|o_i)$.  We didn't include these in Figure 1 because a feedback policy doesn't necessarily always exist, and we were strictly visualizing our model of the *environmental dynamics*.  An example of a situation in which there would be no edges from $o_i$ to $a_i$ would be if the actions were specified *a priori*, for example from some hard-coded action sequence.
>
> 4. The fixed variance was chosen based on a few experiments with different variance values, but we found that it didn't have a huge effect on results.  We will include these experiments in the appendix of the final manuscript version.
>
> [1] Laskin, Michael, Aravind Srinivas, and Pieter Abbeel. "Curl: Contrastive unsupervised representations for reinforcement learning." International conference on machine learning. PMLR, 2020.
>
> [2] Yarats, Denis, Ilya Kostrikov, and Rob Fergus. "Image augmentation is all you need: Regularizing deep reinforcement learning from pixels." International conference on learning representations. 2021.
>
> [3] Yarats, Denis, et al. "Mastering visual continuous control: Improved data-augmented reinforcement learning." arXiv preprint arXiv:2107.09645 (2021).
>
> [5] Laskin, Misha, et al. "Reinforcement learning with augmented data." Advances in neural information processing systems 33 (2020): 19884-19895.
>
> [6] Agarwal, Rishabh, et al. "Contrastive behavioral similarity embeddings for generalization in reinforcement learning." arXiv preprint arXiv:2101.05265 (2021).
>
> [7] Lee, Kimin, et al. "Network randomization: A simple technique for generalization in deep reinforcement learning." arXiv preprint arXiv:1910.05396 (2019).
>
> [8] Nguyen, Tung D., et al. "Temporal predictive coding for model-based planning in latent space." International Conference on Machine Learning. PMLR, 2021.
>
> [9] Zhu, Chuning, et al. "Repo: Resilient model-based reinforcement learning by regularizing posterior predictability." Advances in Neural Information Processing Systems 36 (2023): 32445-32467.
>
> [10] Zhang, Amy, et al. "Learning Invariant Representations for Reinforcement Learning without Reconstruction." International Conference on Learning Representations.
>
> [11] Srivastava, Nitish, et al. "Robust robotic control from pixels using contrastive recurrent state-space models." arXiv preprint arXiv:2112.01163 (2021).

---

### Meta-Review · Area_Chair_J4Q6 · 2025-12-12

**Summary:**

The paper received mixed reviews.
The ratings are 6,2,4,8,4 and all reviewers list weaknesses and have questions/concerns.

The authors provided responses, but none of the reviewers participated in a discussion or provided a post-rebuttal position.

After carefully checking the paper, the reviews, and the authors' responses, the ACs agree with the majority of the reviewers that the paper has issues and is not ready for publication even if the authors provided some responses. The paper requires a major revision and a second review round.

**Reviewer Concerns:**

The paper received mixed reviews.
The ratings are 6,2,4,8,4 and all reviewers list weaknesses and have questions/concerns.

The authors provided responses, but none of the reviewers participated in a discussion or provided a post-rebuttal position.

After carefully checking the paper, the reviews, and the authors' responses, the ACs agree with the majority of the reviewers that the paper has issues and is not ready for publication even if the authors provided some responses. The paper requires a major revision and a second review round.

**Reviewer Scores:**

The paper received mixed reviews.
The ratings are 6,2,4,8,4 and all reviewers list weaknesses and have questions/concerns.

The authors provided responses, but none of the reviewers participated in a discussion or provided a post-rebuttal position.

After carefully checking the paper, the reviews, and the authors' responses, the ACs agree with the majority of the reviewers that the paper has issues and is not ready for publication even if the authors provided some responses. The paper requires a major revision and a second review round.

---

### Decision · Program_Chairs · 2026-01-26

Reject